# A Novel A > G Polymorphism in the Intron 1 of *LCORL* Gene Is Significantly Associated with Hide Weight and Body Size in Dezhou Donkey

**DOI:** 10.3390/ani12192581

**Published:** 2022-09-27

**Authors:** Tianqi Wang, Xiaoyuan Shi, Ziwen Liu, Wei Ren, Xinrui Wang, Bingjian Huang, Xiyan Kou, Huili Liang, Changfa Wang, Wenqiong Chai

**Affiliations:** Liaocheng Research Institute of Donkey High-Efficiency Breeding and Ecological Feeding, Agricultural Science and Engineering School, Liaocheng University, Liaocheng 252059, China

**Keywords:** Dezhou donkey, *LCORL*, body size, hide weight, SNP

## Abstract

**Simple Summary:**

The annual shortage of donkey hides is significant in China. Therefore, to meet the market demand, it is important to improve donkey hide production through molecular breeding. In this study, one SNP was identified in intron 1 of the *LCORL* gene, and polymorphism information content (PIC) was moderate in the population. The association analysis of g.112558859 *A* > *G* with body height, body length, chest circumference and hide weight was conducted and found to be significantly associated with these traits. The hide weight of donkeys with the *GG* genotype was 1.07 Kg heavier than those with *AA* genotype. This provides the foundation for breeding large donkey breeds with high hide production.

**Abstract:**

Several studies have shown the association between the ligand-dependent nuclear receptor compression-like protein (*LCORL*) gene and body size in horses, pigs and donkeys. Based on previous studies, the *LCORL* gene was hypothesized to be associated with growth traits and hide weight in Dezhou donkeys. In this study, we aimed to reveal the variation of the *LCORL* gene in the Dezhou donkey and explore whether the gene is associated with hide weight and body size. In this study, genetic polymorphisms in the *LCORL* gene of the Dezhou donkey were studied using targeted sequencing technology, and single nucleotide polymorphisms (SNPs) of the *LCORL* gene were analyzed for association with hide weight and body size in Dezhou donkeys. The results showed that there was an SNP locus situated in intron 1 of the *LCORL* gene. Association analysis revealed that individuals with the *GG* genotype had significantly higher body height, body length, chest circumference and hide weight than those with the *AA* genotype (*p* < 0.05). Therefore, the g.112558859 *A* > *G* locus can be used as a potential candidate marker affecting body size and hide weight. This study provides the foundation for breeding high-quality donkeys with high hide yield.

## 1. Introduction

The Dezhou donkey is one of the five most important donkey breeds in China [1]. At present, the breeding direction of the Dezhou donkey is to breed high-quality donkeys with high hide and meat yield. In China, a type of gelatin called “donkey gum” or “Ejiao” extracted from donkey hides has medicinal and rejuvenating effects [2]. Since antiquity, Ejiao has been characterized by immunological regulation, anti-oxidation and anti-fatigue, and it has a certain auxiliary effect on the treatment of chronic aplastic anemia [3]. The prices of Ejiao and donkey hides have risen dramatically in the past few decades [2], making donkey hide weight an important economic indicator. The annual shortage of donkey hides is large in China, and Ejiao producers need to import large amounts of donkey hides to meet market demand [2]. Therefore, increasing the yield of donkey hides through molecular breeding has become important. 

There are relatively few studies on donkey hides, and the factors affecting the yield of donkey hides are mainly the thickness and the area of the hides. Wang et al. [4], using transcriptomics analysis, identified *KRT10*, *KRT1*, *CLDN9*, *MHCII* and *MMP-28* as possibly affecting donkey hide thickness. However, a standard method for measuring donkey hide area has not been established. Therefore, hide weight has become a commonly used indicator in measuring donkey hide production. To the best of our knowledge, candidate marker loci for donkey hide weight have not been reported in previous studies. The factors affecting the area of donkey hide include body length, chest circumference and body height. Therefore, screening for mutation loci related to body size and hide weight is valuable for breeding high-quality donkeys with high hide production. 

Body size is a complex trait regulated by various genes and signal pathways. *IGF1* [5], *IGF2* [6], *ACSL1* [7], *ACSL3* [8], *CDKL5* [9] and *TBX3* [10] have been proven to be associated with the body size traits of Dezhou donkeys. For example, the *TBX3* gene was found to be significantly associated with body height, body length and chest circumference in donkeys [10]. In addition, Liu et al. [11] showed that the *TBX3* gene was significantly associated with body size in horses. The *LCORL* gene has been proven to be associated with the body size of livestock in a large number of studies. Rubin et al. [12] used whole-genome resequencing to explore the locus affecting this change and showed that the *LCORL* gene was significantly associated with body length. Graber et al. [13] found that the *LCORL* gene was associated with body height in goats. Tozaki et al. [14] found that the BIEC2-808543 locus, located upstream of the *LCORL* gene, was significantly associated with body height in horses (*p* < 0.05). In donkeys, Shen et al. [15] performed selective scanning, gene annotation, functional enrichment and differential expression analysis on 78 donkeys including 12 breeds of different body sizes and found that the *LCORL* gene was associated with body size. Thus, the *LCORL* gene is considered as a candidate gene that may affect donkey body size and hide weight.

In this study, polymorphisms in the *LCORL* gene of Dezhou donkeys were studied by targeted sequencing, and its association with body size traits and hide weight were investigated. This has laid the foundation for breeding high-quality large donkey breeds with high hide production. 

## 2. Materials and Methods

### 2.1. Moral Statement 

The experimental animals and methods used in this study were approved by the Animal Policy and Welfare Committee of Liaocheng University (no. LC2019-1). The care and use of laboratory animals were in full compliance with local animal welfare laws, guidelines and policies. 

### 2.2. Animals and Phenotypes

A total of 396 blood samples of Dezhou donkeys were collected at a slaughterhouse located in Dezhou, Shandong Province during the wintertime of 2018–2021. The 396 Dezhou donkeys in this study were all males and were 2 years old. The donkeys were from same farm, kept under standard conditions, with the same diet and management conditions [16]. Body size data and hide weight were measured, including body height, body length and chest circumference. Body height, body length and chest circumference were measured in accordance with the National Standard of the People’s Republic of China, “Dezhou Donkey”. Hide weight was weighted immediately after slaughter. Blood samples were collected from the jugular vein using EDTA blood collection tubes and immediately stored in a −20 °C refrigerator [17]. 

### 2.3. DNA Extraction

The 396 genomic DNA samples of Dezhou donkeys were extracted from whole blood with the M5 FlexGen Blood DNA kit (TIANGEN, Beijing, China) [17]. Then, DNA purity (OD_260_/OD_280_) was detected using a spectrophotometer (B500, Metash, Shanghai, China) and the quantity of extracted genomic DNA was detected by 1% agarose gel [18]. Genomic DNA concentrations were measured and working solutions were adjusted to concentrations of 50 ng/µL [19]. 

### 2.4. SNP Detection and Genotyping

The 396 genomic DNA samples were sent to Molbreeding Biotechnology Co., Ltd. (Shijiangzhuang, China) for targeted sequencing. A total of 1902 probes were used in the targeted sequencing, covering 87.29% of the *LCORL* gene with reference sequence of the donkey *LCORL* gene (GenBank accession number: NC_052179.1).

### 2.5. SNPs Validation

SNPs with genotype frequencies greater than 5% were amplified for Sanger sequencing to validate the targeted sequencing results. PCR amplification was performed based on the Dezhou donkey *LCORL* gene sequence (GenBank accession number: NC_052179.1) using primers designed by Primer Premier 5.0 software (PREMIER Biosoft International, San Francisco, CA, USA). The primer sequences are shown in Table 1 for the g.112558859 *A* > *G* locus. The PCR amplification system consisted of 25 μL, including 12.5 μL PCR Mix (Mei5bio, Beijing, China), 8.5 μL ddH_2_O, 1 μL forward primer (Sangon Biotech, Shanghai, China), 1 μL reverse primer (Sangon Biotech, Shanghai, China) and 2 μL genomic DNA. The cycling parameters were as follows: 95 °C for 5 min, 30 cycles of denaturing at 95 °C for 30 s, annealing at 58 °C for 30 s and extension at 72 °C for 30 s, with a final extension at 72 °C for 10 min [17]. The PCR products were tested for specificity using 2% agarose gel, and those with specificity and correct fragment size were randomly screened 1 sample per genotype, and the samples were sent to BGI Genomics Co., Ltd. (Shenzhen, China) for Sanger sequencing. The results were analyzed using Chromas software (Version V2.6.5, Technelysium Pty Ltd., Queensland, Australia).

### 2.6. Statistical Analysis 

SNPs with genotype frequencies less than 5% were removed from the targeted sequencing results. Genotype frequency, allele frequency and Hardy Weinberg equilibrium (HWE) were calculated. The GDIcall online calculator (http://www.msrcall.com/gdicall.aspx, accessed on 1 July 2022.) was used to calculate genetic parameters, including polymorphic information content (*PIC*), homozygosity (*Ho*), heterozygosity (*He*) and effective allele numbers (*Ne*) [20]. Associations between SNP locus genotypes and phenotypes (body height, body length, chest circumference and hide weight) in Dezhou donkeys were analyzed using a general linear model in SPSS 26.0 (Statistical Product and Service Solutions, Version 26.0 Edition, IBM, Armonk, NY, USA) [21]. The analytical model is as follows: Yij=μ+ai+eij
where *Y* is the individual phenotypic measure, *μ* denotes the mean value of body size trait and hide weight, *a* indicates the fixed factor genotype and *e* stands for random error. Age, sex, sampling season and rearing environment were consistent and their effects were not considered in this model. Least squares means with standard deviations were used for different genotypes, body size traits and hide weight. *p* < 0.05 indicates that the difference is statistically significant. The substitution effects (*α*), dominance effects (*d*) and additive effects (*a*) of alleles on body size traits and hide weight in Dezhou donkeys were calculated as follows: *a* = (*GG* − *AA*)/2, *d* = *AG* − (*GG* + *AA*)/2 and α = a + d (p − q), where *GG* and *AA* represent the mean values of individuals with homozygous genotypes, *AG* represents the mean values of individuals with heterozygous genotypes, and p and q represent the gene frequencies of *G* and *A* alleles, respectively. 

## 3. Results and Discussion

### 3.1. Genetic Polymorphism of LCORL Gene in Dezhou Donkey

A total of 346 SNPs were detected by targeted sequencing, of which 9 SNPs were located upstream, 12 SNPs were situated in the exon region, 325 SNPs were located in the intron region and no SNPs were detected in the downstream region (Appendix A). Genotype frequencies of the majority of SNPs were less than 5%, except for one SNP, g.112558859 *A* > *G* located in intron 1 (Figure 1). Samples of the three genotypes were stochastically selected for Sanger sequencing validation, and these results were in agreement with the results of the targeted sequencing. Figure A1 and Figure A2 show the PCR product reference sequences and mutation sequences, respectively. Figure A3 shows the g.112558859 *A* > *G* locus sequencing peak diagram. Three genotypes were detected at the g.112558859 *A* > *G* locus, which were *AA*, *AG* and *GG* genotypes. Similarly, Shen et al. [15] identified 11 nonsynonymous loci on the *LCORL* gene. 

### 3.2. Genetic Parameters Analysis

The sequence variation frequencies and population genetic parameters for g.112558859 *A* > *G* locus in the *LCORL* gene of the Dezhou donkey are shown in Table 2. The frequencies of the *GG*, *AG* and *AA* genotypes were 0.4419, 0.4015 and 0.1566, respectively. The number of *GG* genotype individuals was the largest, indicating that the mutated genotype at this site was the main genotype in the population. The frequencies of *G* and *A* alleles were 0.6225 and 0.3775, respectively, indicating that mutated alleles were dominant in this population. This locus was in Hardy Weinberg equilibrium, indicating that the population was large and interbred freely [22]. *Ho*, *He* and *Ne* of g.112558859 *A* > *G* locus were 0.5300, 0.4700 and 1.8867, respectively. *PIC* was 0.3595, showing moderate polymorphism (0.25 < *PIC* < 0.5). This result indicates that the population has a high level of polymorphism and the locus is genetically stable [23]. However, considering that the animals used in our study were from the same farm, our results have some limitations, and further expansion of the sample size is needed. 

### 3.3. Association Analysis of LCORL Gene g.112558859 A > G Locus with Hide Weight and Body Size 

The association analysis of the *LCORL* gene g.112558859 *A* > *G* locus polymorphism with hide weight and body size was performed, and the results are shown in Table 3 and Figure 2. Individuals with the *GG* genotype had significantly higher body height, body length, chest circumference and hide weight than those with the *AA* genotype (*p* < 0.05). g.112558859 *A* > *G* locus *GG* genotype individuals had 1.94 cm higher body height than *AA* genotype individuals, 2.35 cm longer body length than *AA* genotype individuals and 2.24 cm higher chest circumference than AA genotype individuals. Similarly, the *LCORL* gene was associated with body height, body length and body size in horses [14], pigs [12] and donkeys [15]. Some research has shown that the *LCORL* gene was significantly associated with carcass weight in cattle [24,25,26]. In our study, even though there was no significant difference in carcass weight (*p* > 0.05) between donkeys with *GG* and *AA* genotypes, the trend showed that donkeys with the *GG* genotype had higher carcass weight than those with the *AA* genotype. The reason could be the sample size limitation; further research is likely needed, with an expansion of the sample size. 

In addition to growth traits, hide weight is also an important indicator for donkey breeding. The hide weights of donkeys with these genotypes were significant different (*p* < 0.05), with donkeys of the *GG* genotype having the highest hide weight; the hide weight of donkeys with the *AA* genotype was 1.07 Kg lighter (Table 2). On the contrary, Wang et al. [4] reported that the *LCORL* gene was not included among the genes associated with donkey hide thickness. The present study shows that the *LCORL* gene g.112558859 *A* > *G* locus increased the hide area by increasing the body height, body length and chest circumference, which in turn led to an increase in hide weight. In the practical application of donkey farms, the molecular markers can be detected after the foal is born, and the donkeys with low hide weight can be phased out.

The g.112558859 *A* > *G* locus of the *LCORL* gene is located in the intron 1 region. Compared to SNPs located in exons, SNPs located in introns are also highly functional [27]. This may be due to mutant sites affecting the binding of splicing factors that form new transcripts, which ultimately affects gene expression. Further studies are needed to determine whether g.112558859 *A* > *G* affects splicing factor binding sites. Guo et al. [28] identified a novel transcript of the bovine SPEF2 gene, which may be caused by the SNP g.11043 *C* > *T* located in intron 1, and altered the splicing factor binding protein SC35 binding to the target sequence; SNP g.11043 *C* > *T* was found to be significantly associated with sperm malformation rate. Chang et al. [6] found that SNPs (g.291322 *C* > *T* and g.281766 *G* > *A*) located in the *IGF2* intron region were significantly associated with rump height and body length in female Dezhou donkeys (*p* < 0.05). Therefore, SNPs located in introns can also be potential molecular markers for donkey breeding. The functions of the g.112558859 *A* > *G* locus still need to be further verified, such as gene knockout technology and transcriptome sequencing.

### 3.4. The Additive Effect, Dominant Effect and Substitution Effect

The additive effect, dominant effect and substitution effect of the *G* allele on the *C* allele are shown in Table 2. The additive effect showed increasing hide weight, carcass weight, body height, body length and chest circumference by 0.54 kg, 2.13 kg, 0.97 cm, 1.18 cm and 1.12 cm, respectively. Additive effects can be stably inherited to offspring [29]. The dominant effect increased hide weight, carcass weight, body height, body length and chest circumference by 0.10 kg, 0.92 kg, 0.63 cm, 0.61 cm and 0.51 cm, respectively. The substitution effect increased hide weight, carcass weight, body height, body length and chest circumference by 0.56 kg, 2.36 kg, 1.12 cm, 1.32 cm and 1.24 cm, respectively. This indicates that the *G* allele may have a positive effect on these traits.

## 4. Conclusions

Improving body size and hide weight by molecular breeding has important application value. In this study, the g.112558859 *A* > *G* locus was identified in the intron 1 region of the *LCORL* gene, and this locus was significantly associated with body height, body length, chest circumference and hide weight (*p* < 0.05). This suggests that g.112558859 *A* > *G* locus can be used as a potential candidate marker for breeding Dezhou donkeys, but further functional validation is needed. These findings provide the foundation for breeding large donkey breeds with high hide production.

## Figures and Tables

**Figure 1 animals-12-02581-f001:**
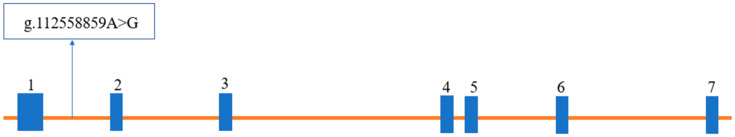
Schematic representation of the *LCORL* gene and localization of identified mutation sites. The blue rectangle represents the exon region and the orange rectangle represents the intronic region.

**Figure 2 animals-12-02581-f002:**
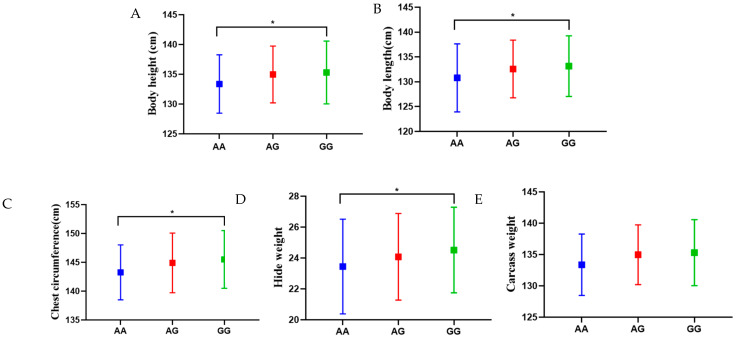
Effect of different genotypes at the g.112558859 *A* > *G* locus on traits. * *p* < 0.05.

**Table 1 animals-12-02581-t001:** Used PCR amplification primers in g.112558859 *A* > *G* locus of Dezhou donkey.

Primer	Location	Primer Sequences (5′–3′)	Tm (°C)	Product Size (bp)
1	Intron 1	F:CCCTGATTACTCTTTCTTTG	58	600
		R: CCTTTGGTTGGCTTGATGAAT		

**Table 2 animals-12-02581-t002:** Sequence variation frequencies and population indexes for the g.112558859 *A* > *G* locus in the *LCORL* gene of the Dezhou donkey.

	Samples	Genotypic Frequencies			Allelic Frequencies		HWE	Ho	He	Ne	PIC
		GG	AG	AA	G	A					
g.112558859 *A* > *G*	396	0.4015 (159)	0.4419 (175)	0.1566 (62)	0.6225	0.3775	0.2350	0.5300	0.4700	1.8867	0.3595

**Note:** HWE: Hardy Weinberg equilibrium; Ho: homozygosity; He: heterozygosity; Ne: effective allele numbers; PIC: polymorphic information content. PIC < 0.25, low polymorphism; 0.25 < PIC < 0.5, intermediate polymorphism; PIC > 0.5, high polymorphism.

**Table 3 animals-12-02581-t003:** Association between the g.112558859 *A* > *G* locus within the *LCORL* gene and growth traits and hide weight of the Dezhou donkey.

SNP	Genotypes	Individual Number	Hide Weight	Carcass Weight	Body Height	Body Length	Chest Circumference
			(kg)	(kg)	(cm)	(cm)	(cm)
g.112558859 *A* > *G*	*AA*/62	62	23.45 ± 3.07 b	148.38 ± 16.31	133.37 ± 4.90 b	130.80 ± 6.84 b	143.27 ± 4.77 b
	*AG*/175	175	24.08 ± 2.80	151.43 ± 19.65	134.97 ± 4.77	132.58 ± 5.80	144.90 ± 5.17
	*GG*/159	159	24.52 ± 2.77 a	152.64 ± 19.59	135.31 ± 5.27 a	133.15 ± 6.11 a	145.51 ± 5.01 a
	*p*-value		0.038	0.333	0.033	0.037	0.013
*α*			0.56	2.36	1.12	1.32	1.24
*a*			0.54	2.13	0.97	1.18	1.12
*d*			0.10	0.92	0.63	0.61	0.51

**Note:** Means with different small within the same row are significantly different (*p* < 0.05). *α* = gene substitution effects; *a* = gene additive effects; *d* = gene dominance effects.

## Data Availability

The data presented in this study are available on request from the corresponding author.

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
