# Peer review of "A Novel A > G Polymorphism in the Intron 1 of LCORL Gene Is Significantly Associated with Hide Weight and Body Size in Dezhou Donkey"

_animals, 2022, doi:10.3390/ani12192581_

Round 1

Reviewer 1 Report

The manuscript “A novel A > G polymorphism in the intron 1 of LCORL gene is significantly associated with hide weight and body size in Dezhou donkey” aims at screening for the variation of LCORL gene linked to hide weight and body size using SNP genotype data of 396 Dezhou donkey individuals, and find the g.112558859 A>G locus was significantly associated with body height, body length, chest circumference and hide weight. The study found the g.112558859 A>G locus may contribute to breeding large donkey breeds with high hide production. However, there are some questions to be noticed.

1, first P>0.05 is far from significant, or this locus did not perform well in body size conformation.

2, As reported from (Wang, 2021), one SNP in TBX3 gene is under association with the height of donkey, Dezhou donkey, the same breed as this study. TBX3 is also found to play an important role in horse height (Liu, 2022). They did not cite or even further discuss about this.

3, There are 1902 target probs designed for target sequencing, while only one SNP (maf >0.05) was filtered. While the gene LCORL is about 150Kb, SNP is too much sparse in donkey genome, which is not credible.

3, in line 169-170, the allele frequencies are 0.44/0.4/0.15, it is of important to show the distribution of each trait associated with different genotypes, as well as the significance levels. And how much body size variation can be explained by this locus?

4, what’s more, only genotyping and association analysis is not enough to conclude if it is a genetic marker for height, it’s functional role should be stressed and validated.

Here are some minor revisions:

1. The English language of the manuscript is not sufficient quality, and the manuscript needs serious revision.

2. In “2.5 SNPs validation”, Please incorporate Table S1 content into the manuscript text, it does not need a separate table.

3. Line 151, where is table S2? It is not found in manuscript.

Author Response

Response to Reviewer 1 Comments

Point 1: first P>0.05 is far from significant, or this locus did not perform well in body size conformation.

Response 1: Thanks for your suggestions. we have already corrected this error.

Point 2: As reported from (Wang, 2021), one SNP in TBX3 gene is under association with the height of donkey, Dezhou donkey, the same breed as this study. TBX3 is also found to play an important role in horse height (Liu, 2022). They did not cite or even further discuss about this.

Response 2: Thanks for your suggestions. it has been modified as suggested. Content is added to the line 60-63. 

Point 3: There are 1902 target probs designed for target sequencing, while only one SNP (maf >0.05) was filtered. While the gene LCORL is about 150Kb, SNP is too much sparse in donkey genome, which is not credible.

Response 3: Thanks for your suggestions. This may be due to the fact that our sample size needs to be further expanded, or it may be due to the fact that our populations are all from the same slaughterhouse and there is inbreeding within the population. We have added Table S2, which shows the sequencing results of 396 samples of LCORL gene SNPs. As shown in Table S2, the number of mutant genotypes at the other loci was small. Furthermore, 56 SNPs were found to have only the wild-type genotype and the remaining loci were not detected during manuscript revision. The loci in this case have been removed as shown in the figure below. Therefore, a total of 346 SNPs were detected by targeted sequencing, of which 9 SNPs were located upstream, 12 SNPs were located in the exon region, 325 SNPs were located in the intron region, and SNPs were not detected in the downstream region.

Point 4: in line 169-170, the allele frequencies are 0.44/0.4/0.15, it is of important to show the distribution of each trait associated with different genotypes, as well as the significance levels. And how much body size variation can be explained by this locus?

Response: Thanks for your suggestions. Content is added to the line 193-195. Significance levels have been increased in Table 2. g. 112558859 A>G locus GG genotype individuals had 1.94 cm higher body height than AA genotype individuals, 2.35 cm longer body length than AA genotype individuals, and 2.24 cm higher chest circumference than AA genotype individuals. The hide weight of donkeys with these genotypes were significant different (P < 0.05) in which the GG genotype has the highest hide weight, and the AA genotype was 1.07 Kg lighter than it.

Point 5: what’s more, only genotyping and association analysis is not enough to conclude if it is a genetic marker for height, it’s functional role should be stressed and validated.

Response: Thanks for your suggestions. Content is added to the line 231-233 and line250. The function of g. 112558859 A>G locus still needs to be further verified, such as gene knockout technology and transcriptome sequencing. Our study suggests that g.112558859 A>G locus can be used as a potential candidate marker for breeding Dezhou donkeys, but further functional validation is needed.

Here are some minor revisions:

Point 1: The English language of the manuscript is not sufficient quality, and the manuscript needs serious revision.

Response: Thanks for your suggestions, we have tried our best to correct the English error.

Point 2: In “2.5 SNPs validation”, Please incorporate Table S1 content into the manuscript text, it does not need a separate table.

Response: Thanks for your suggestions, it has been modified as suggested.

Point 3: Line 151, where is table S2? It is not found in manuscript.

Response: Thanks for your suggestions. Table S1 and Table S2 are in the Supplementary materials.

Reviewer 2 Report

The candidate gene study presented is clear presented. In principle, it would be better to conduct a genome-wide association study using e.g. SNP genotypes, as the trait under investigation is clearly polygenic (quantitative) in nature. Nevertheless, the selection of the LCORL gene seems logical and plausible based on the findings in other domestic animal species.

It would be good to list the most recent studies on body size in the species mentioned and to cite them correctly in the introduction, e.g. a similar study on goats was recently published in the journal Animal Genetics.

When presenting the results, it would be better to show the measurements summarised in Table 2 in relation to the three genotypes additionally in the form of boxplots, then the described (significant) differences would be better visible. Furthermore, the P values of the respective contrasts between the genotype groups must be added to Table 2.

The presented sequence variants should be described with their genomic position in the current goat reference assembly, so that it is clear in the long term where the variant is located.

Sections B and C of Figure 1 should be moved to the appendix, as the information presented seems insignificant at this point.

Finally, the discussion and the conclusion lack a conceivable functional explanation of the effect of this intron variant on the expression of the affected gene and thus a hypothesis as to why the mutation has the observed effect on the traits studied.

Author Response

Response to Reviewer 2 Comments

Point 1: It would be good to list the most recent studies on body size in the species mentioned and to cite them correctly in the introduction, e.g. a similar study on goats was recently published in the journal Animal Genetics.

Response 1: Thanks for your suggestions. Content is added to the line 66-67.

Point 2: When presenting the results, it would be better to show the measurements summarised in Table 2 in relation to the three genotypes additionally in the form of boxplots, then the described (significant) differences would be better visible. Furthermore, the P values of the respective contrasts between the genotype groups must be added to Table 2.

Response 2: Thanks for your suggestions. The influence of different genotypes at g. 112558859A> G locus on traits has been graphically shown in Figure 2. The P values of the respective contrasts between the genotype groups have been increased in Table 2.

Point 3: The presented sequence variants should be described with their genomic position in the current goat reference assembly, so that it is clear in the long term where the variant is located.

Response 3: Thanks for your suggestions. The SNP nomenclature used is the position in the genome, e.g. g. 112558859 A>G, 112558859 is the position of the locus in the Dezhou donkey genome. PCR product sequences have been added in the appendix section.

Point 4: Sections B and C of Figure 1 should be moved to the appendix, as the information presented seems insignificant at this point.

Response 4: Thanks for your suggestions, it has been modified as suggested.

Point 5: Finally, the discussion and the conclusion lack a conceivable functional explanation of the effect of this intron variant on the expression of the affected gene and thus a hypothesis as to why the mutation has the observed effect on the traits studied.

Response 5: Thanks for your suggestions, it has been modified as suggested. Content is added to the line 218-221.